# Insights into the Adolescent Cystic Fibrosis Airway Microbiome Using Shotgun Metagenomics

**DOI:** 10.3390/ijms25073893

**Published:** 2024-03-31

**Authors:** Gillian McDermott, Aaron Walsh, Fiona Crispie, Susanna Frost, Peter Greally, Paul D. Cotter, Orla O’Sullivan, Julie Renwick

**Affiliations:** 1Trinity Centre for Health Science, Clinical Microbiology Department, School of Medicine, Faculty of Health Science, Trinity College Dublin, Tallaght University Hospital, D24 NR0A Dublin, Ireland; gmcderm@tcd.ie; 2Teagasc Food Research Centre, Moorepark, Fermoy, P61 C996 Co Cork, Ireland; awalsh12@tcd.ie (A.W.); fiona.crispie@teagasc.ie (F.C.); paul.cotter@teagasc.ie (P.D.C.); orla.osullivan@teagasc.ie (O.O.); 3APC Microbiome Ireland, University College Cork, T12 R229 Co Cork, Ireland; 4Tallaght University Hospital, Tallaght, D24 NR0 Dublin, Irelandpeter.greally@tuh.ie (P.G.); 5Hermitage Medical Clinic, Lucan, D20 W722 Dublin, Ireland

**Keywords:** cystic fibrosis, whole-genome sequencing, metagenomics, microbiome, resistance, resistome, sequence types, persistence

## Abstract

Cystic fibrosis (CF) is an inherited genetic disorder which manifests primarily in airway disease. Recent advances in molecular technologies have unearthed the diverse polymicrobial nature of the CF airway. Numerous studies have characterised the genus-level composition of this airway community using targeted 16S rDNA sequencing. Here, we employed whole-genome shotgun metagenomics to provide a more comprehensive understanding of the early CF airway microbiome. We collected 48 sputum samples from 11 adolescents and children with CF over a 12-month period and performed shotgun metagenomics on the Illumina NextSeq platform. We carried out functional and taxonomic analysis of the lung microbiome at the species and strain levels. Correlations between microbial diversity measures and independent demographic and clinical variables were performed. Shotgun metagenomics detected a greater diversity of bacteria than culture-based methods. A large proportion of the top 25 most-dominant species were anaerobes. Samples dominated by *Staphylococcus aureus* and *Prevotella melaninogenica* had significantly higher microbiome diversity, while no CF pathogen was associated with reduced microbial diversity. There was a diverse resistome present in all samples in this study, with 57.8% agreement between shotgun metagenomics and culture-based methods for detection of resistance. Pathogenic sequence types (STs) of *S. aureus*, *Pseudomonas aeruginosa*, *Haemophilus influenzae* and *Stenotrophomonas maltophilia* were observed to persist in young CF patients, while STs of *S. aureus* were both persistent and shared between patients. This study provides new insight into the temporal changes in strain level composition of the microbiome and the landscape of the resistome in young people with CF. Shotgun metagenomics could provide a very useful one-stop assay for detecting pathogens, emergence of resistance and conversion to persistent colonisation in early CF disease.

## 1. Introduction

Cystic Fibrosis (CF) is the most common inherited life-shortening disease in Caucasians. CF is caused by mutations in the CF Transmembrane Conductance Regulator (CFTR) gene [1], resulting in dysfunctional osmotic regulation across epithelial membranes. Viscous airway mucous and reduced mucociliary clearance create the ideal environment for microorganisms to persist. Although it is a multiorgan disease, up to 95% of morbidity and mortality is caused by pulmonary infections [2]. The most common bacteria isolated from the airways of people with CF are *Staphylococcus aureus*, *Pseudomonas aeruginosa*, *Haemophilus influenzae*, *Stenotrophomonas maltophilia* and *Acinetobacter baumannii*. *P. aeruginosa* is the most common cause of exacerbation in CF, and up to 80% of adults with CF are colonised [3]. Chronic *P. aeruginosa* colonisation has been independently linked to increased mortality, morbidity and reduced lung function [4,5].

While the clinical significance of *P. aeruginosa* in CF is undeniable, the past two decades has seen the introduction of advanced next-generation sequencing techniques that have revolutionised CF airway microbiology. It is now well established that the CF airway is host to a diverse community of bacteria [6], fungi [7] and viruses [8]. Many of these microbes were never before associated with the CF airway, and a large portion of the CF airway microbiota is now known to be populated by anaerobes [9,10]. While these studies have advanced CF microbiology enormously, most of them have used targeted 16S rDNA gene sequencing, which, in general, cannot be used to profile microbiota below the genus-level [11,12,13,14]. In contrast, whole-genome shotgun metagenomics has been shown to have improved taxonomic resolution [15], while it also has the added benefit of providing insights into the potential functions encoded by the microbiome, including virulence and antimicrobial resistance. Here, we employed shotgun metagenomics to provide in-depth characterization of the CF airway microbiome. We monitored the persistence of bacteria in patients over time at the species and strain levels, while we also established the complete resistance gene repertoire (i.e., the “resistome”) within the CF airway microbiome. Finally, we associated microbiome parameters to clinical and demographic factors. 

## 2. Results

### 2.1. Clinical, Demographic and Microbiological Data

Shotgun metagenomics was performed on 48 samples collected from 11 patients. On average, four consecutive samples were collected from a patient (range; 1–9) over the 12-month period. The sample set comprised 12 *P. aeruginosa* culture positive samples from six patients and 36 *P. aeruginosa* culture negative samples from five patients. There was no significant difference in age and *P. aeruginosa* positivity between the male and female groups (Table 1). An even distribution of ∆F508 homozygosity and heterozygosity was also recorded in these cohorts. Mean body mass index (BMI) z-score and mean forced expiratory volume in 1 s (FEV_1_) % predicted were both significantly higher in males than in females (*p* = 0.0369 and *p* = 0.0014, respectively). Such gender disparities in CF have been previously reported [16].

### 2.2. The CF Airway Microbiome Changes within Patients over Time

Following quality control, there was an average of 1,776,458 ± 3,084,513 reads per sample (185,162 to 19,123,414) (Appendix A). An average of 12 bacterial species per sample (1–85 species/sample) were detected by MetaPhlAn2, which was higher than the average number of bacterial species detected by culture of 2.5 (1–8 species/sample) (Appendix A). The composition of the microbiome was found to vary within individuals over time, and alpha diversity was found to change over short periods (Figure 1).

Among the top 25 most abundant species were those frequently associated with CF lung samples, namely, *P. aeruginosa*, *S. aureus*, *H. influenzae*, *H. parainfluenzae* and *S. maltophilia*, alongside emerging CF bacteria *Gemella (Ge) haemolysans*, *Ge. sanguinis*, an unclassified *Granulicatella (Gr)*, *Rothia aeria*, *R. dentocariosa*, *R. mucilaginosa*, *Porphyromonas sp oral taxon 279*, *Prevotella melaninogenica*, *Veillonella atypica*, *V. parvula* and an unclassified *Veillonella sp* (Figure 2A). A variety of non-*pneumoniae* streptococci were identified among the most abundant species, including *S. salivarius*, *S. infantis*, *S. parasanguinis* and *S. sanguinis*. An unclassified *Capnocytophaga* and *Neisseria* species were also among the most abundant species.

All the samples were dominated by at least one of the following species: *S. aureus*, *S. maltophilia*, *H. influenzae*, *P. aeruginosa*, an unclassified *Pseudomonas* species and *P. melaninogenica*. Samples positive for *S. aureus* and *P. melaninogenica* had significantly higher Shannon’s alpha diversity measures than non-colonised samples (Kruskal-Wallis with Dunn’s multiple comparison; *p* = 0.001 and *p* = 0.012, respectively) (Figure 2B). No other dominant species was linked to differences in alpha diversity. A significant negative correlation was detected between the relative abundances of *S. maltophilia* and *P. aeruginosa* (Spearman’s test, *p* = −0.459, *p* = 0.007).

Culture-based analysis detected only three bacterial species dominating samples: *S. aureus*, *P. aeruginosa* and *S. maltophilia.* There was 75% agreement between culture *(n* = 35) and shotgun metagenomics (*n* = 37) for *S. aureus* positivity. There were four samples detected by culture but undetected by shotgun metagenomics and seven samples detected by shotgun metagenomics that were undetected by culture. Similarly, there was high agreement between shotgun metagenomics (*n* = 12) and culture (*n* = 15) for detecting *P. aeruginosa* (85%) and *S. maltophilia* (83%). Shotgun metagenomics detected *P. aeruginosa* in two samples that were undetected by shotgun metagenomics, and five samples were undetected by culture but detected by shotgun metagenomics. Seven samples were undetected for *S. maltophilia* by culture but detected by shotgun metagenomics, and only one sample was detected by culture but undetected by shotgun metagenomics. None of the emerging bacteria were detected in the samples by culture.

### 2.3. Diversity of the CF Airway Microbiome Was Not Associated with Lung Function (FEV_1_), Sex or Antibiotics

A linear mixed-effects model accounting for repeated measures within subjects was used to test if alpha diversity was associated with the following patient variables: antibiotic usage, lung function and sex. Although alpha diversity decreased with antibiotic usage, the association was non-significant (*p* = 0.375). Similarly, although alpha diversity decreased with deteriorating lung function, the association was non-significant (*p* = 0.375). Finally, although alpha diversity was lower in females, the association was not significant (*p* = 0.086) (Appendix A).

Multi-dimensional scaling indicated that intra-individual variation was lower than inter-individual variation, which was confirmed by measuring the Bray–Curtis distances between samples (Appendix A). PERMANOVA was used to determine if patient variables were associated with beta diversity. Again, there were no significant associations between beta-diversity and antibiotic usage (*p* = 0.232), lung function (*p* = 0.239) or sex (*p* = 0.607).

### 2.4. The CF Airway Microbiome Harbours Mechanisms of Resistance Not Detected by Culture

To investigate the resistome, reads were aligned to the MEGARes antimicrobial resistance (AMR) database [17] using Bowtie 2 [18]. The identity and abundance of AMR genes was determined using ResistomeAnalyzer (https://github.com/cdeanj/resistomeanalyzer) (accessed on 28 February 2024). Multi-drug resistance (MDR) (e.g., porins and efflux pumps) and genes conferring resistance towards aminoglycosides were common to almost all samples from all patients in our study (Figure 3A). Additionally, genes predicted to encode resistance to tetracyclines, sulfonamides, rifampicin, phenicol, lipopeptides, fosfomycin, fluoroquinolones, elfamycins, ß-lactams and aminocoumarins were present in most CF airway microbiomes (Figure 3A). Genes conferring trimethoprim, glycopeptide, fusidanes and cationic antimicrobial peptide resistance were rare in these microbiomes.

Streptogramins, elfamycins, cationic antimicrobial peptides, aminocoumarins and lipopeptides are not routinely tested for in diagnostic laboratories and so were not detected using culture methods. Additionally, detection of multi-drug resistance (MDR) refers to the presence of genes for porins and efflux pumps which may be present but not necessarily confer antibiotic resistance. Not considering these antimicrobials and resistance mechanisms, there was only 57.8% agreement between culture and shotgun metagenomics for the detection of resistance (Figure 3B). 

The most common resistance mechanisms identified by culture were those towards aminoglycosides, ß-lactams, lincosamides and sulfonamides. There was only 52% and 47.6% agreement between culture and shotgun metagenomics for ß-lactam and aminoglycoside resistance detection, respectively (Figure 3C). Only four samples were positive for aminoglycoside resistance by culture and not shotgun metagenomics, while 16 samples were positive by shotgun metagenomics and negative by culture. Nine samples were ß-lactam resistant by culture but negative by shotgun metagenomics, and nine samples were positive for ß-lactam resistance by shotgun metagenomics but negative by culture. Genes for resistance to tetracyclines, phenicol, lipopeptides, macrolides, streptogramines, fosfomycins, elfamycins and aminocoumarins were abundant in the CF airway microbiome yet only detected by shotgun metagenomics (Figure 3C). In general, there was a considerable difference between the resistance detected by culture/Vitek and that detected by shotgun metagenomics. 

### 2.5. Patients Colonised with Pathogenic Strain Types Tended to Be Colonised with the Same Strain Type over Time

In order to assess the power of shotgun metagenomics to discriminate to the strain level and monitor pathogenic strains, we employed MetaMLST (version 1.2.2). Eleven pathogenic sequence types (ST) were identified in 23 samples from nine patients, with ≥98.52% confidence (Table 2). *S. aureus* STs appeared most frequently with a total of eight different STs detected in our samples. ST5 was the most prevalent *S. aureus* ST, and all were shown to have methicillin resistance genes by PanPhlAn analysis. Patients 7 and 20 were persistently colonised with *S. aureus* ST5, while patient 9 was colonised with different *S. aureus* STs over time. In some samples, pathogenic STs corresponding to more than one species were identified; for example, *S. aureus* and *S. maltophilia* STs were both detected in sample 7_J (Table 2). All patient 9 samples were co-colonised with two species of pathogenic STs. ST4 was detected at three time points in patient 8. Although two other sequence types (ST100001 and ST100002) were detected, both overlapped with ST4 considerably: ST100001 shared 5/7 alleles with ST4, while ST100002 shared 6/7 alleles with ST4. 

MetaMLST has been shown to have a higher false negative rate in the detection of pathogenic STs when compared with other tools [19]; therefore, we employed the PanPhlAn tool to complement the MetaMLST findings and identify further pathogenic strains. PCA was performed on the PanPhlAn gene families table (Figure 4A). Patient 9 was persistently colonised with the same *H. influenzae* ST in three consecutive samples. Patients 7 and 14 were colonised with a similar *H. influenzae* ST, and patients 3 and 22 were also colonised with a highly similar *H. influenzae* ST. Patients 3, 11 and 22 had distinct pathogenic strains of *P. aeruginosa* (PA01, Nhmuc & DK1, SCV and B136-33). Patient 11 had the same pathogenic *P. aeruginosa* Nhmuc strain in two consecutive samples (GCF_001900265, 81.2% identity) and a very similar DK1 strain in a third sample (GCF_900069025, 81.4% identity). Five patients’ samples (7, 8, 9, 20 and 27) had the same pathogenic *S. aureus* FDAARGOS_16 strain (GCF_001019355, 81.5–95.6% identity) in consecutive samples, indicating patients potentially sharing strains or a common source. Four samples from patient 11 harboured the same pathogenic FDAARGOS_29 strain of *S. aureus* (GCF_001018965, 75.7–85.6% identity), and two samples from patient 2 harboured the same *S. aureus* NRS 143 strain (GCF_001018975, 84.1–85.8% identity). Overall, a large proportion (~64% of samples; 7/11 patients) of patient samples had pathogenic *S. aureus* strains. Patient 8 showed the same pathogenic *S. maltophilia* K279a strain in all consecutive samples (GCF_000072485, 68.3–73.1% identity). Patients 12 and 7 harboured a very similar pathogenic strain of *S. maltophilia*. Again, there was evidence of co-colonisation with more than one species of pathogenic ST. Patient 9 was co-colonised with pathogenic STs of *H. influenzae* and *S. aureus*, and patient 11 was co-colonised with pathogenic STs of *P. aeruginosa* and *S. aureus* over multiple samples. Samples from patients 7 and 8 had pathogenic strains of both *S. aureus* and *S. maltophilia.*

The StrainPhlAn tool was then used to construct phylogenetic trees of the pathogenic STs detected in the samples (Figure 4B). Both the PanPhlAn tool and StrainPhlAn tool indicated that the same STs were present in individual patients over time. Patients 7, 8, 9 and 20 had highly related *S. aureus* strains, and patients 7 and 9 were persistently colonised with this strain. Mirroring the PanPhlAn results, StrainPhlAn showed patient 8 to be persistently colonised with a highly related pathogenic ST of *S. maltophilia*. The *S. maltophilia* strains colonising patients 7 and 12 were shown to be highly genetically similar and could indicate patients sharing strains. Patient 9 was persistently colonised with a highly related *H. influenzae* ST. StrainPhlAn confirmed that patients 7 and 14 were harbouring highly related strains. The StrainPhlAn tool only detected pathogenic *P. aeruginosa* STs in two samples and so we could not construct a tree for *P. aeruginosa*. 

## 3. Discussion

CF microbiome studies to date have largely performed targeted 16S rDNA gene sequencing and, therefore, are primarily limited to genus-level analysis. More recently, a small number of studies have employed whole-metagenome sequencing to provide more in-depth analysis of the airway microbiome in CF [20,21,22,23]. Here, we performed shotgun metagenomics on 48 sputum samples from 11 children and adolescents with CF. Our patient cohort, although small, is well defined and includes sequential samples from a subset of patients. We were able to characterise the species and even strain-level composition of the CF airway microbiome. Furthermore, we have explored the diverse resistome of the CF airway and identified persistent colonisation with pathogenic strains of common CF bacteria. 

Pathogen identification and the prescribing of antimicrobials form the cornerstone of CF care. Diagnostic laboratories still rely heavily on traditional culture-based methods, which have high rates of false negatives and potential for false positives [24]. The CF airway is now known to harbour a polymicrobial community of microorganisms [25,26], and many of these lesser known or “emerging” organisms have been shown to impact CF disease [27,28]. In this study, we found the well-known CF bacteria *P. aeruginosa*, *S. aureus*, *H. influenzae*, *H. parainfluenzae* and *S. maltophilia* to be some of the most abundant species across most samples, mirroring decades of culture data. A number of *Streptococcus* spp were also among the most abundant species in our samples and have been detected in previous CF microbiome studies [6,29]. We identified two *Gemella* species, a *Prevotella* species, three *Rothia* species and three *Veillonella* species among the most abundant bacteria present in our samples, and none of these bacteria were detected by culture. *Ge. haemolysans*, *Ge. sanguinis*, *Pr. melaninogenica*, *Porphyromonas*, *V. atypica*, *V. parvula*, *St. infantis*, *St. mitis oralis pneumoniae*, *St. parasanguinis*, *St. sanguinis* and *St. salivarius* have all previously been described in the context of the CF airway [25,26,30,31]. While *R. dentocariosa* and *R. mucilaginosa* have been associated with the CF airway [26], *R. aeria* has only before been associated with the non-CF bronchiectasis airway [32]. Other bacteria detected in this study but only scarcely reported in CF microbiome studies were *Capnocytophaga* [25,33,34] and *Granulicatella* [31]. The importance of some of these newly emerging anaerobes in CF disease has been highlighted. In a study of 68 paired baseline and exacerbation sputum samples collected from 28 patients with CF, *Gemella* abundance was reported to be the most discriminative genus between baseline and exacerbation samples [27]. Tunney et al. reported that *Prevotella* isolates from the CF airway can produce ß-lactamases shielding ß-lactam-sensitive *P. aeruginosa* from antibiotic killing [28]. Other studies are now establishing the influence of these emerging species on the virulence and resistance of common CF pathogens such as *P. aeruginosa* [35,36,37,38,39]. These findings show the potential for these emerging species, alone or as part of the microbial community, to impact CF disease. 

A higher microbiome complexity was detected in *S. aureus*-dominant samples in this study. One CF registry study including 28,042 patients spanning 9 years reported that meticillin-sensitive *S. aureus* (MSSA) was negatively associated with subsequent *P. aeruginosa* colonisation and suggested a positive role for *S. aureus* by potentially delaying *P. aeruginosa*-mediated lung disease progression [40]. A number of studies have now been published reporting that *S. aureus* and *P. aeruginosa* interact with each other [41,42]. The use of anti-staphylococcal prophylaxis in paediatric CF patients may make patients vulnerable to colonisation with more detrimental pathogens such as *P. aeruginosa* [43]. Flucloxacillin prophylaxis for children with CF is widely debated. It is currently recommended from infancy in the United Kingdom [44], advised against in the United States [45], while in Ireland there is no consensus. Studies attempting to determine the benefit of anti-staphylococcal prophylaxis have reported contradicting outcomes [43,46]. One trial reported that children treated with *S. aureus* prophylaxis were more likely to become positive for *P. aeruginosa* [43]. Conversely, other work has proven effective clearance of *S. aureus* and improved lung function upon anti-staphylococcal treatment [47]. Four patients in our study were taking flucloxacillin at one sampling time, and two patients were taking flucloxacillin at two sampling times (eight samples in total). *P. melaninogenica* was also associated with increased diversity in the CF airway microbiome, and, to our knowledge, this is the first report of this in the literature. Muhlebach et al. (2018) found that anaerobes were associated with milder disease in a large multi-centre study [9], and previous work by our group has revealed that a number of anaerobes, including *Prevotella spp*, were significantly more abundant in the non-CF airway [48]. Together with our findings, it is clear that anaerobes have a significant role to play in the CF airway, with most studies indicating an association with health. 

Microbiome diversity was not associated with lung function in this study. Others have described this association in older patient populations [49,50,51,52,53]. The reported impacts of antibiotics on the CF airway microbiome have been contradictory. Some studies have identified antibiotics as the primary cause of reduced microbiome complexity [49], while others have reported antibiotics to have little impact on the overall diversity of the CF microbiome [40,54,55]. Here, we were unable to detect any correlation between antibiotic usage and microbiome diversity of the airway in young people with CF. However, this may be due to the small patient cohort. 

As we performed shotgun metagenomics, we were able to probe the genomes for antimicrobial resistance genes and compare this to culture-based resistance detection. The CF airway microbiome has a diverse repertoire of resistance genes, and there was 57.8% agreement between culture and shotgun metagenomics detection. Feigelman et al. (2017) performed paired-end sequencing on the Illumina HiSeq platform on spontaneous/induced sputum from six adult CF patients, four patients with COPD and seven healthy controls (three smokers/four non-smokers) [21]. They reported good agreement between culture-dependent antimicrobial resistance testing and their sequencing annotated resistance genes. In this study, there was 46% agreement between shotgun metagenomics and culture for detection of ß-lactam resistance. There was equal reporting of false negatives by both shotgun metagenomics and culture for detection of ß-lactam resistance. It has been demonstrated that extended-spectrum ß-lactamases produced by lesser-known anaerobes of the CF microbiome, such as *Prevotella*, can shield more pathogenic species from ß-lactam activity, providing passive resistance [28]. In a recent study, 96% of CF isolates of *P. melanogenica* were ß-lactamase producers [56], indicating that uncultured bacteria may be a reservoir for resistance in the CF airway microbiome. There is also mounting evidence that polymicrobial interactions in the host can alter resistance, which would be undetectable in the pure culture lab environment [39,41,57]. Interpreting traditional culture and antibiotic susceptibility data in patients with CF is challenging and does not always correlate with response to treatment. Choosing antibiotic regimens purely based on the results of these traditional tests has been shown to produce poor clinical outcomes [58,59]. CF clinicians often prescribe antimicrobial regimens which make little sense in the context of the printed laboratory result but which are associated with a clinical improvement in the patient, leading most to opine that current culture-based resistance testing is not adequate to inform treatment choices in many CF patients. This view is shared by the Laboratory Standards for Processing Microbiological Samples from People with Cystic Fibrosis (2010) (UK CF trust Laboratory Standards working group) [60]. 

Recurrent or chronic infections are a feature of CF airway disease and are associated with poor therapeutic responses, reduced lung function and reduced life expectancy. Persistent infection with *P. aeruginosa* in particular is a significant disease milestone. In this study, patients were persistently colonised with pathogenic STs of *P. aeruginosa*, *S. aureus*, *H. influenzae* and *S. maltophilia*. All of these bacteria have previously been shown to evolve and adapt in the CF airway [61,62,63,64]. Bacci et al. (2020) performed whole-metagenomics shotgun sequencing on 79 samples from 22 adolescents and adults with CF and found seven patients were persistently colonised by pathogenic STs of *P. aeruginosa* [65]. In contrast, we only detected persistent *P. aeruginosa* in one of 11 patients. However, this is a younger cohort where chronic *P. aeruginosa* colonisation would be less common. Persistent colonisation with MRSA in CF has been linked with a more rapid decline in lung function than in CF patients persistently colonised with MSSA [66], and we detected persistent colonisation by the methicillin resistant pathogenic *S. aureus* ST5 in several patients. *S. maltophilia* is an inherently multi-drug resistant organism, and persistent infection has been associated with increased risk of pulmonary exacerbation, lung transplantation and death [67]. Here, we observed persistent colonisation with *S. maltophilia* ST4 in one patient, which has been shown to be a common ST among CF patients [68]. *H. influenzae* is considered a commensal of the nasopharynx, and, while it is a common coloniser of the CF airway in childhood, its role in CF disease is controversial as there is limited evidence of the impact of chronic *H. influenzae* colonisation on clinical course. A study by Faliu F. et al. (2021) has shown that chronic infection with non-typable *H. influenzae* can cause inflammation in in-vitro and in-vivo mouse models, suggesting that *H. influenzae* can contribute to inflammatory burden in people with chronic lung diseases [69]. We have shown that pathogenic STs of *H. influenzae* can persist in the early CF airway, and the impact on early lung disease should be considered. As persistent infections are associated with increased AMR and reduced therapeutic options, shotgun metagenomics could be employed to flag early signs of persistence, and earlier interventions could be planned to reduce or delay chronic infection. 

Five patients’ samples had the same pathogenic *S. aureus* ST, indicating potential patient-to-patient transmission or acquisition from a common source. Ankrum and Hall (2017) performed whole-genome sequencing on 311 *S. aureus* isolates collected from 115 CF patients and found that only siblings shared *S. aureus* STs and therefore attributed transmission events to the home environment [70]. None of the five patients in our study had familial relationships. There was limited evidence of other transmission events in this patient population. 

This study provided a detailed analysis of the airway microbiome in 11 children and adolescents with CF. While this is a small patient cohort, we were able to resolve species and even strain-level dynamics of the microbiome and identify persistent colonisation of several pathogenic STs. The full collective of AMR genes was detected and revealed that culture does not thoroughly represent the diverse repertoire of resistance mechanisms present in the CF airway microbiome. The correlation of resistance detected by shotgun metagenomics and response to treatment merits further investigation to try to understand the clinical significance of shotgun metagenomics resistance testing and its potential role in influencing treatment strategies in patients with chronic lung infections. Shotgun metagenomics studies have provided a fresh insight into the CF airway microbiome. Dmirejeva et al. (2021) demonstrated how whole-genome sequencing data can be employed as an additional parameter to current clinical measures to monitor longitudinal sublineage pathogen dynamics and improve understanding of individual patient disease course [71]. In contemplating such an approach, the inability of shotgun metagenomics, and indeed all molecular techniques, to distinguish between live and dead microorganisms would need to be thoroughly considered. Shotgun metagenomics could potentially be used as a one-stop assay to identify colonising species, determine resistance profiles, track transmission events and identify patients who are chronically colonised by pathogenic strains. 

## 4. Materials and Methods

### 4.1. Patient Cohort

Sputum samples (*n* = 48) from 11 children and adolescents with CF were collected based on clinical criteria during suspected pulmonary exacerbations and in convalescence following therapy as part of standard care. Consecutive samples from each patient were collected between 1 and 5 months apart. Demographic data such as age, gender and CFTR mutation were collected alongside clinical data including body mass index (BMI) z score, forced expiratory volume percentage predicted in 1 s (FEV_1_) and antibiotics prescribed in the last 2 months. For lung deterioration comparisons, patients were categorised into mild deteriorating lung function (72–89% FEV_1_), moderate deterioration (48–59% FEV_1_) and severe deterioration (28–37% FEV_1_), as previously published [72]. The entire dataset has been fully anonymised with no keyholder, and ethics approval for the use of these samples in research was granted by the Tallaght University Hospital St James’s Hospital joint ethics committee (reference code: 2019-09 List 35).

### 4.2. Sample Collection and Processing

Sputum samples were homogenised with 1:1 Sputasol (Oxoid Ltd., Hampshire, UK). Samples were then split into two equal volumes with one volume used for culture and the other for shotgun metagenomics. Initially, both tubes were centrifuged at 9500× *g* at 4 °C for 10 min. The pellet of one tube was resuspended in 1 mL RNA*later*^®^ (Invitrogen by Thermo Fisher Scientific, Dublin, Ireland), incubated at 4 °C overnight and then stored at −80 °C for future shotgun metagenomics. The pellet of the second tube was resuspended in Phosphate Buffered Saline (PBS) (Sigma-Aldrich, Merck Life Science Ltd., Wicklow, Ireland) to a total volume of 1 mL and used for culture on the day of collection.

### 4.3. Culture

Reference strains (*P. aeruginosa*–PA27853, *H. influenzae*–HI8468, *S. aureus*–SA25923, *S. maltophilia*–SM17666 and *A. baumannii* (MALDI ToF confirmed clinical isolate)) were maintained on Colombia blood agar plates (Oxoid Ltd., Hampshire, UK). A serial dilution of all sputum samples was prepared and plated onto Colombia sheep blood agar, chocolate agar (Oxoid Ltd., Hampshire, UK), cetrimide agar (Fannin Ltd., Dublin, Ireland), *S. aureus* ID agar (SAID agar) (Biomerieux UK Ltd., Basingstoke, UK), CHROMID^®^
*S. aureus* Elite agar (Biomerieux UK Ltd., Basingstoke, UK), Chocolate agar with bacitracin (Fannin Ltd., Dublin, Ireland), Vancomycin, Imipenem & Amphotericin B agar (VIA agar) [73], CHROMagar™ Acinetobacter and CHROMagar^TM^ Orientation (CHROMagar^TM^, Paris, France). Following incubation in the appropriate environmental conditions in triplicate (Appendix A), the number of Colony Forming Units (CFU) per millilitre of original sputum was calculated. Ultimately, bacterial species identification was determined using commercial matrix-assisted laser desorption/ionization time-of-flight mass spectrometry (MALDI-TOF MS) (Biomerieux UK Ltd., Basingstoke, UK), and antimicrobial sensitivity was determined employing the VITEK 2.0 (Biomerieux UK Ltd., Basingstoke, UK) system.

### 4.4. DNA Extraction and Human DNA Depletion

Sample pellets were resuspended in 200 μL molecular grade water and bead beaten using 0.3 g sterile acid washed glass beads (Sigma-Aldrich, Merck Life Science Ltd., Wicklow, Ireland) for 180 s at 5.5 m/s [74]. A 200 µL volume of binding buffer and 40 µL proteinase K were added and incubated at 70 °C for 10 min. Following incubation, Roche high pure PCR template preparation kit (Roche diagnostics Ltd., West Sussex, UK) instructions were followed. For each batch of extractions, a negative extraction control was performed where the protocol was carried out from start to finish on 200µL molecular grade water. DNA was quantified on the Qubit fluorometer (Life Technologies, Thermo Fisher Scientific, Bleiswijk, Netherlands) employing the Qubit^TM^ dsDNA High Sensitivity and Broad Range assay kit (Invitrogen by Thermo Fisher Scientific, Dublin, Ireland). Then, 16S rDNA gene amplification was performed on negative extraction controls. Microbial DNA was enriched using the NEBNext^®^ Microbiome DNA Enrichment Kit (New England Biolabs Ltd., Hitchin, UK) [75] as per the manufacturer’s instructions. Enriched DNA then underwent ethanol precipitation and was stored at −20 °C.

### 4.5. Shotgun Metagenomic Sequencing

Whole-metagenome shotgun libraries were prepared in accordance with the Nextera XT DNA Library Preparation Guide from Illumina, with the exception that the tagmentation time was increased to 7 min. After indexing and clean-up of the PCR products, as described in the protocol, each sample was run on an Agilent bioanalyser high sensitivity chip (Agilent Technologies Ireland Ltd., Cork, Ireland) to determine the size range of the fragments obtained. The concentration of the samples was also determined at this point using a Qubit^TM^ dsDNA High Sensitivity and Broad Range assay kit (Invitrogen by Thermo Fisher Scientific, Dublin, Ireland). Samples were then pooled equimolarly, and the final concentration of the pooled library was determined by quantitative PCR using the Kapa Library Quantification kit for Illumina (Roche diagnostics Ltd., West Sussex, UK). The pooled library was then sequenced on the Illumina NextSeq 500 using the 2 × 150 High Output kit according to standard Illumina sequencing protocols.

### 4.6. Bioinformatic Analysis of Sequencing Reads

Human reads were removed from the raw shotgun metagenomic reads using NCBI Best Match Tagger (BMTagger v1.1.0). Next, fastq files were converted to unaligned bam files using SAMtools (v1.13) [76], and duplicate reads were subsequently removed using Picard Tools (v2.7.1). Low quality reads were then removed using the trimBWAstyle.usingBam.pl script from the Bioinformatics Core at the UC Davis Genome Center (https://github.com/genome/genome/blob/master/lib/perl/Genome/Site/TGI/Hmp/HmpSraProcess/trimBWAstyle.usingBam.pl) (accessed on 28 February 2024). Specifically, reads were filtered to 105 bp, while those with a quality score less than Q30 were discarded. The resulting fastq files were then converted to fasta files using the fq2fa option from IDBA-UD (v1.1.1) [77]. Fasta files have been deposited in the European Nucleotide Archive (ENA) (Accession: PRJEB52482).

MetaPhlAn2 (v2.6.0) [78] was used to perform species-level analysis, while strain-level analysis was performed using Pangenome-based phylogenomic analysis (PanPhlAn v1.2) [79]. PanPhlAn aligns analysis reads against the pangenome database, which allows functional characterisation of strains (http://segatalab.cibio.unitn.it/tools/panphlan/) (accessed on 28 February 2024). Reconstruction of multi-locus sequence types (MLST) was carried out using MetaMLST [80] (version 1.2.2). MetaMLST is a means of reconstructing MLST loci of microorganisms present in the microbiome of a sample from metagenomics data (https://github.com/SegataLab/metamlst) (accessed on 28 February 2024). The heatmap generated for the top 25 most abundant species was constructed using hclust2 (https://github.com/SegataLab/hclust2) (accessed on 28 February 2024), and samples were clustered based on Euclidean distance. Resistome analysis was performed by aligning reads against the MEGARes antimicrobial resistance (AMR) database [17] using Bowtie 2 (v2.2.5) [18]. The identity and abundance of AMR genes was determined using ResistomeAnalyzer (https://github.com/cdeanj/resistomeanalyzer) (accessed on 28 February 2024).

### 4.7. Statistical Analyses

The statistical analyses applied to the resulting sequencing data were performed in R-3.2.2. The vegan package (version 2.3.0) was used for alpha diversity analysis, Bray–Curtis based multidimensional scaling (MDS) analysis and permutational analysis of variance analysis (PERMANOVA) [81]. A linear mixed effects model was implemented in the lmerTest package (version 3.1.3) to test if alpha diversity was associated with clinical variables. Similarly, the MaAslin2 package [82] (version 1.14.1), which implements linear mixed effects models for microbiome studies, was used to test if the abundances of species were associated with clinical variables. The following formula was used across each approach: microbiome ~ antibiotics + FEV1 + sex + (1|patient). We accounted for subject-specific effects as follows: the PERMANOVA model was stratified by patient, and the linear mixed effects models included patient as a random effect. Correlation analysis was performed using the rcorr function from the Hmisc package (version 5.1.0) [83]. Specifically, the Spearman method was used to correlate the relative abundance of *P. aeruginosa* to the relative abundances of all other species detected with MetaPhlAn2. The data were visualised using the ggplot2 package (version 2.2.1) [84]. The code used for the statistical analyses is available on GitHub: https://github.com/ (accessed on 28 February 2024).

## Figures and Tables

**Figure 1 ijms-25-03893-f001:**
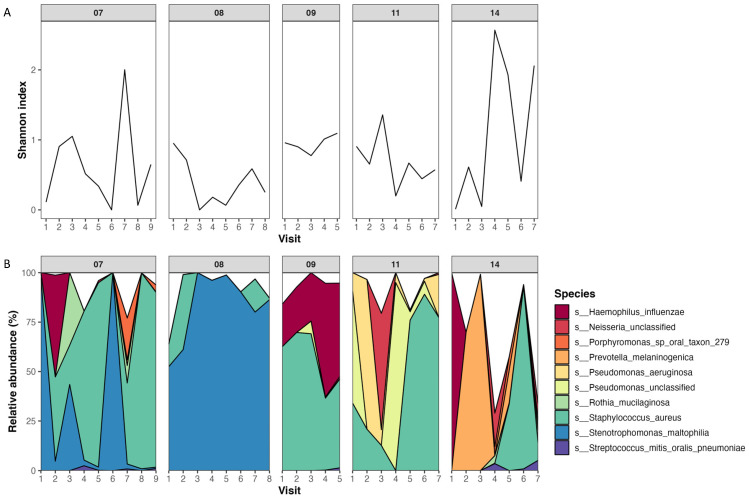
Temporal changes in the lung microbiome of CF patients. The plots are faceted by patient, and only patients with ≥5 samples are shown. (**A**) Alpha diversity of the lung microbiome at each visit as measured by the Shannon index. (**B**) Species relative abundances at each visit. Only the top 10 species are shown, which were determined by averaging species’ relative abundances across all samples.

**Figure 2 ijms-25-03893-f002:**
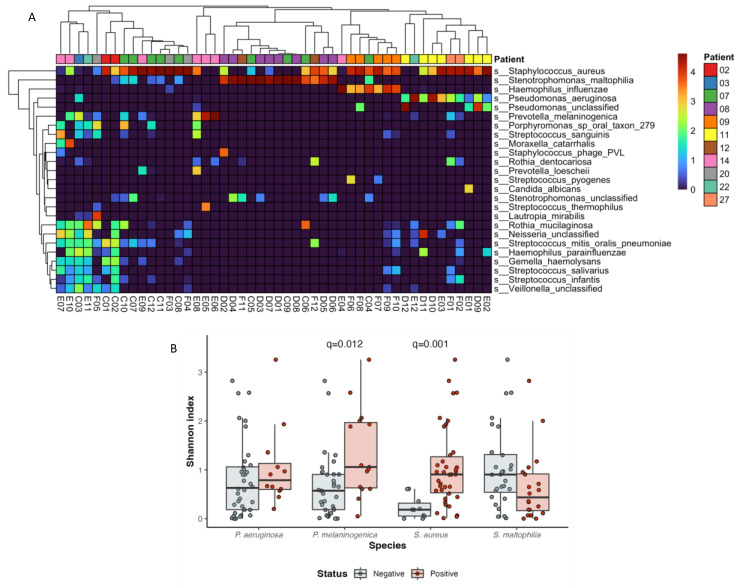
(**A**) Heatmap and dendrogram of relative abundances of the 25 most abundant species across all patient samples. The tiles on the heatmap are coloured by the log relative abundance of the species per sample, while the colour bar indicates the patient from which the sample was taken. (**B**) Comparison of alpha-diversity (Shannon’s diversity indices) between pathogen-dominant and pathogen-negative samples. Comparison of groups by Kruskal–Wallis with Dunn’s multiple comparison.

**Figure 3 ijms-25-03893-f003:**
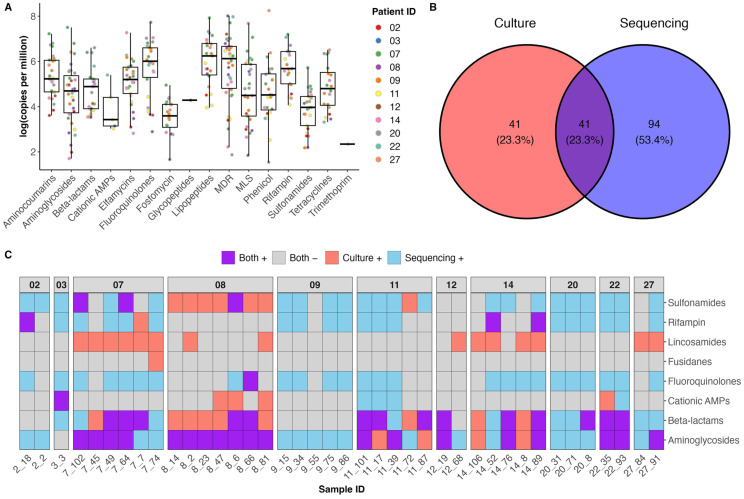
The presence of antibiotic resistance genes (ARGs) and antibiotic resistant phenotypes in the CF airway microbiome. (**A**) The summed abundance of ARG classes detected by ResistomeAnalyzer, expressed as Log copies per million. (**B**) The per sample and (**C**) overall concordance between the metagenomic detection of ARGs and culture-confirmed antibiotic susceptibility. MLS = macrolide, lincosamides and streptogramines.

**Figure 4 ijms-25-03893-f004:**
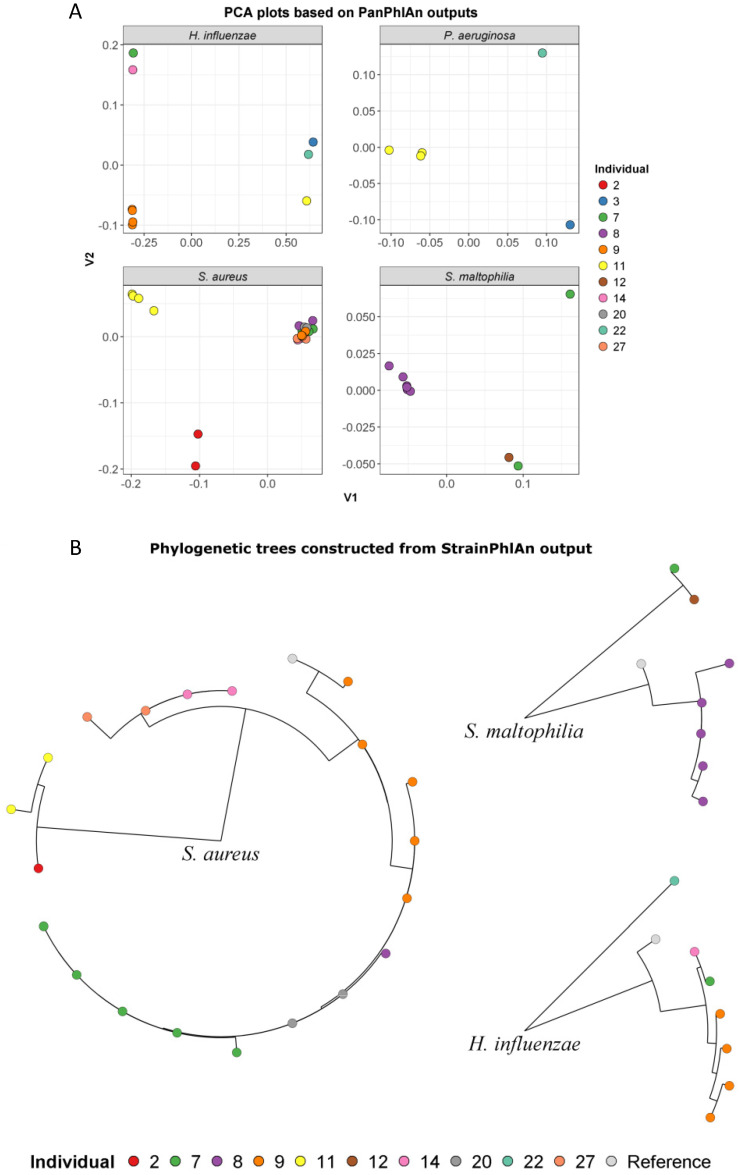
Dissimilarity in genetic composition of pathogenic strain types of *P. aeruginosa*, *S. aureus*, *S. maltophilia* and *H. influenzae* based on PanPhlAn output. (**A**) PCA plots of genetic distance of pathogenic strains. (**B**) Phylogenetic trees (StrainPhlAn tool) of the pathogenic STs detected in samples. Samples are colour coded according to patient key.

**Table 1 ijms-25-03893-t001:** Patient Demographic and Clinical details.

		Number (%; Range; StDev)
Demographics	Gender (F)	5/11 (45.5%)
	Median Age	16 (6–19)
Clinical measures	Average BMI z-score	−0.489 (±1.524)
	Average FEV_1_% predicted	63% (±24)
CFTR mutation	∆F508/∆F508	9/11 (81.8%)
	∆F508/other	2/11 (18.2%)
Microbial culture positivity	*S. aureus*	10/11 (90.9%)
	*P. aeruginosa*	5/11 (45.5%)
	*S. maltophilia*	4/11 (36.4%)
	*Candida* spp	5/11 (45.5%)
	*A. fumigatis*	3/11 (27.3%)
	Other microorganisms ^a^	4/11 (36.4%)
	Normal flora ^b^	6/11 (54.5%)
Antimicrobials in prior 2 months ^c^	Azithromycin	6/11 (54.5%)
	Flucloxacillin	6/11 (54.5%)
	Co-trimoxazole	5/11 (45.5%)
	Colomycin	4/11 (36.4%)
	Amoxacillin/clavulanic acid	4/11 (36.4%)
	Tobramycin	3/11 (27.3%)
	Doxycycline	2/11 (18.2%)
	Meropenem	2/11 (18.2%)
	Linezolid	2/11 (18.2%)
	Other antimicrobials ^d^	10/11 (90.9%)
	Median no. of antimicrobials	2 (0–7)

^a^ Other microorganisms are those present in only one patient and <50% of samples from that patient (*Chryseobacterium gleum*, *Klebsiella oxytoca*, *Enterobacter cloacae*, *Pseudomonas putida*, *Streptococcus multivorum*, *Mycobacterium abscessus*, *Pseudomonas stutzeri*, *Aspergillus tereus*, *Staphylococcus epidermidis*, *Achromobacter xyloxidans*, *Achromobacter denitrificans*, *Streptococcus pyogenes*). ^b^ Normal flora refers to alpha haemolytic Streptococci that are not *S. pneumoniae* present in a respiratory sample. ^c^ Antimicrobials refers to the number of patients on that antimicrobial in the prior 2 months. ^d^ Other antimicrobials are those prescribed to ≤1 patient at one or more time points (Clarithromycin, Erythromycin, Colistin, Ciprofloxacin, Amikacin, Ofloxacin, Aztreonam, Ceftazidime, Trimethoprim, Itraconazole).

**Table 2 ijms-25-03893-t002:** Pathogenic sequence types detected using MetaMLST.

Species	Patient	Sample Number	Sequence Type (ST)
*H. influenzae*	7	7_A	100001
*S. aureus*	7	7_G	5
*S. aureus*	7	7_I	100001
*S. aureus*	7	7_J	5
*S. maltophilia*	7	7_J	5
*S. aureus*	7	7_L	5
*S. maltophilia*	8	8_B	4
*S. maltophilia*	8	8_D	4
*S. maltophilia*	8	8_F	100001
*S. maltophilia*	8	8_G	4
*S. maltophilia*	8	8_J	100002
*S. aureus*	9	9_A	100003
*S. pyogenes*	9	9_A	100001
*H. influenzae*	9	9_B	100002
*S. aureus*	9	9_B	100004
*H. influenzae*	9	9_D	100003
*S. aureus*	9	9_D	100005
*H. influenzae*	9	9_F	100004
*S. aureus*	9	9_F	5
*S. aureus*	11	11_G	100002
*S. maltophilia*	12	12_A	5
*H. influenzae*	14	14_A	368
*S. aureus*	14	14_E	15
*S. aureus*	14	14_F	15
*S. aureus*	20	20_A	5
*S. aureus*	20	20_B	5
*P. aeruginosa*	22	22_B	313
*S. aureus*	27	27_B	845

## Data Availability

Fasta files have been deposited in the European Nucleotide Archive (ENA) (Accession: PRJEB52482).

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
