# Peer review of "Insights into the Adolescent Cystic Fibrosis Airway Microbiome Using Shotgun Metagenomics"

_ijms, 2024, doi:10.3390/ijms25073893_

Round 1

Reviewer 1 Report

Comments and Suggestions for Authors

This manuscript delivers novel and interestig results on the time course of pathogen invasion in CF, which might be futher developed by clinicians to adjust and time  their treatment regimes to delay arrival of more detrimental microbes. Thus the manuscript is of high value, both to the non-medical and medical community.

The manuscript may benefit from adding -either in introduction or in the beginning of the results section - a section on a more general introduction of  statistic analyses employed in the study. The authors present a  mansucript the merit of which resides in a careful analysis of a complex data set. Readers not familiar in detail  with the program suites employed to analyze and vizualize the data sets might greatly benefit  from a paragraph explaining these.

This is a very fine paper and it is respectfully noted that the authors acknowledge the children/ young adults suffering from CF and their parents for participating in this study.

Author Response

Thank you for your kind comments about our work and this paper. We have addressed your requests for additional descriptions of the statistical methods used as per below.

Section 4.7 details the statistical methods used in the study and we have now updated this to include more information. We have also supplied a link to GitHub where all the coding used for the analysis in this study has been made accessible (https://github.com/aaron-breathnach/cf).

Reviewer 2 Report

Comments and Suggestions for Authors

Authors present an in-vitro study on 48 sputum samples from children and adolescents with cystic fibrosis (CF) taken over a period of 12 months (Pseudomonas aeruginosa positive and negative cultures) with employment of  whole genome shotgun metagenomics on the ilumina NextSeq platform to provide a more comprehensive understanding of the early CF airway microbiome.Shotgun metagenomics detected greater diversity of bacteria than culture-based  methods, of 25 dominant species largely anaerobes,  diverse  resistome present in all samples in this study with 57.8% agreement between shotgun metagenomics and culture-based methods for detection of resistance. Pathogenic sequence types  of S. aureus,  P. aeruginosa, Haemophilus influenzae and Stenotrophomonas maltophilia were observed to persist in9 young CF patients while STs of S. aureus were both persistent and shared between patients.

Low number of patients is a major drawback of the study. I suggest to include a table with all patients and their current clinical condition - is there a correlation between the severity of the disease and the microbiome?

Comments on the Quality of English Language

Moderate changes. 

Author Response

While there are low patient numbers we have been able to follow several of these patients over consecutive months resulting in 48 samples. Metagenomic studies are expensive and we made the decision to monitor a small number of patients more closely rather than sample ~50 patients at one time point. We chose not to overload our Illumina NextSeq 500 run to provide deeper sequencing of the 48 samples from 11 patients.

We have been transparent about the issues with interpreting data from small cohort studies and deliberately draw reserved conclusions (lines 318-321; lines 382-384), Our patient cohort, although small, is well defined and includes sequential samples from a subset of patients and “Here we were unable to detect any correlation between antibiotic usage and microbiome diversity of the airway in young people with CF however this may be due to the small patient cohort.” We have also now included reference to the cohort size being small in the final conclusion paragraph of the discussion to reinforce this (line 447).

Respectfully, we would prefer to keep Table 1 as it currently is for GDPR reasons. Table 1 and the corresponding paragraph (section 2.1) provides a clinical and demographic overview of the patient population which is commonly seen in the field (Doi: 10.3389/fcimb.2020.00173; 10.1513/AnnalsATS.201211-107OC). We note the reviewer requests “current clinical conditions” of all patients. This data has been fully anonymised therefore we are unable to follow up with current details of their disease status.

We performed analysis to see if clinical measures of disease severity, specifically Forced Expiratory Volume in 1 second (FEV1) (lines 475-477) could be linked to microbiome alpha and beta diversity measures and were unable to find such associations. These results are detailed in section 2.3 and supplementary figure S3. This is discussed in lines 377-384 of the discussion section.
